# Genetic Engineering and Innovative Cultivation Strategies for Enhancing the Lutein Production in Microalgae

**DOI:** 10.3390/md22080329

**Published:** 2024-07-23

**Authors:** Bert Coleman, Elke Vereecke, Katrijn Van Laere, Lucie Novoveska, Johan Robbens

**Affiliations:** 1Aquatic Environment and Quality, Cell Blue Biotech and Food Integrity, Flanders Research Institute for Agriculture, Fisheries and Food (ILVO), Jacobsenstraat 1, 8400 Ostend, Belgium; bert.coleman@ilvo.vlaanderen.be (B.C.);; 2Plant Sciences Unit, Flanders Research Institute for Agriculture, Fisheries and Food (ILVO), Caritasstraat 39, 9090 Melle, Belgium; 3Department of Plant Biotechnology and Bioinformatics, Ghent University, Technologiepark 71, 9052 Zwijnaarde, Belgium; 4Center for Plant Systems Biology, Flemish Institute for Biotechnology (VIB), Technologiepark 71, 9052 Zwijnaarde, Belgium; 5ScotBio, Livingston EH54 5FD, UK

**Keywords:** microalgae, carotenoids, lutein, cultivation strategies, genetic engineering

## Abstract

Carotenoids, with their diverse biological activities and potential pharmaceutical applications, have garnered significant attention as essential nutraceuticals. Microalgae, as natural producers of these bioactive compounds, offer a promising avenue for sustainable and cost-effective carotenoid production. Despite the ability to cultivate microalgae for its high-value carotenoids with health benefits, only astaxanthin and β-carotene are produced on a commercial scale by *Haematococcus pluvialis* and *Dunaliella salina,* respectively. This review explores recent advancements in genetic engineering and cultivation strategies to enhance the production of lutein by microalgae. Techniques such as random mutagenesis, genetic engineering, including CRISPR technology and multi-omics approaches, are discussed in detail for their impact on improving lutein production. Innovative cultivation strategies are compared, highlighting their advantages and challenges. The paper concludes by identifying future research directions, challenges, and proposing strategies for the continued advancement of cost-effective and genetically engineered microalgal carotenoids for pharmaceutical applications.

## 1. Lutein as One of the Important Carotenoids

Carotenoids are a class of fat-soluble pigments that include carotenes (β-carotene, lycopene) and xanthophylls (astaxanthin, lutein, fucoxanthin). Carotenes are hydrocarbon carotenoids, while xanthophylls are the oxygenated versions of carotenes. Humans cannot synthesize carotenoids and must obtain them through their diet. These compounds are mainly produced by plants, fungi, and microorganisms, with a total of 1204 natural carotenoids identified [1]. They primarily absorb light at wavelengths of 400–550 nm. In photosynthetic organisms, carotenoids play a crucial role in protecting photosynthetic organisms against photodamage and supporting their oxygenic photosynthesis [2].

Carotenoids have many health benefits, as listed in Table 1. Astaxanthin is the strongest antioxidant among carotenoids, with significantly stronger antioxidant activity and free radical inhibitory activity than vitamin E, β-carotene, and lutein [3]. The synthetic version is used in aquaculture to give fish (i.e., salmon) and crustaceans their pinkish color. β-Carotene is well-known as a precursor to vitamin A and is also used as natural orange-yellow color in the food industry [4]. Lutein is widely used as an antioxidant, food coloring agent, and nutritional supplement in cosmetics, food, health products, and medicine [5]. Other carotenoids, including zeaxanthin and fucoxanthin, also offer health benefits (Table 1).

The global carotenoid market was valued at approximately USD 1.8 billion in 2021 and is projected to reach USD 2 billion by 2026 [8]. The astaxanthin market is projected to surpass USD 800 million by 2026. The β-carotene market is expected to be worth USD 620 million by 2026, with a CAGR of 3.8% from 2018 to 2026 [9]. The lutein market is projected to reach USD 358 million by the end of 2024 [8]. The global fucoxanthin market reached a valuation of USD 95 million in 2020 and is projected to attain a value of USD 113.5 million by 2026 [10]. Other carotenoids like zeaxanthin, lycopene, and canthaxanthin are also gaining support and increasing market interest due to their health benefits. The market shares by application are dominated by animal feed (34.8%), followed by food and beverages (26.1%), dietary supplements (23.5%), pharmaceuticals (9.2%), and cosmetics (6.5%) [8]. The global carotenoid market is poised for substantial growth, driven by increasing demand across various applications and the rising awareness of their health benefits.

In the evolving market of carotenoids, there is a significant contrast between natural and synthetic sources, each with its own cost implications, safety considerations, and market share dynamics. Carotenoids produced by chemical synthesis can be up to three times cheaper than those derived from natural sources. Consequently, natural carotenoids only capture 10–20% of the market share due to high production costs [5]. However, this ratio is changing as concerns about safety and environmental impact grow on synthetic carotenoids and the higher health benefits of natural carotenoids become clear [11]. Synthetic carotenoids are predominantly used in applications such as animal feed and as colorants, whereas natural carotenoids are preferred for use in medicine and food supplements [11]. Another major reason for the preference towards natural carotenoids is their significant differences in forms and bioactivity compared to synthetic sources, as natural carotenoids are usually complex mixtures of various isomers, whereas synthetic carotenoids typically consist of a single form [11]. The market value of synthetic carotenoids is relatively low, ranging from USD 250 to USD 2000 per kilogram, whereas natural carotenoids from plant sources range from USD 350 to USD 7500 per kilogram [12]. This price difference reflects the higher consumer demand and perceived benefits of natural carotenoids over their synthetic counterparts. 

## 2. Microalgae as Producers of Lutein and Other Carotenoids

Microalgae are emerging as significant producers of valuable carotenoids, with biosynthesis occurring in their chloroplasts. Microalgae are a diverse group of eukaryotic aquatic microorganisms that can thrive in fresh, brackish, or marine water, with over 200,000 species classified into various phylogenetic groups [13]. Depending on the species, microalgae can be cultivated photoautotrophically, using light as an energy source to assimilate CO_2_ for biomass production, or heterotrophically, utilizing an organic carbon source for energy. Some species can grow mixotrophically, using both light and organic carbon sources. Photoautotrophic microalgae can be cultivated in open raceway ponds (ORPs) or closed photobioreactors (PBRs) [13,14]. These systems can be placed outdoors using natural sunlight, indoors with artificial lighting, or in greenhouses with natural sunlight. ORP systems are characterized by lower investment and operational costs but are more susceptible to biological contamination compared to closed PBRs [15]. In contrast, PBRs offer easier control over cultivation parameters and higher biomass productivity. Heterotrophic microalgae are cultivated in fermenters (closed systems) where an organic carbon source is added. These fermenters can achieve higher cell densities (25–125 g DW/L) compared to PBRs (0.5 g DW/L) [13]. Heterotrophic cultivation in well-controlled bioreactors is gaining commercial attention for pigment production due to its ability to overcome challenges related to CO_2_ and light supply, as well as contamination and land requirements in open autotrophic systems [16]. Microalgae growth can be managed in various modes: batch, fed-batch, semi-continuous, and continuous modes [13]. In batch mode, microalgae are grown with all nutrients provided at the start, continuing until nutrients deplete. It is simple to operate, which also reduces the risk of contamination, but it generally results in lower biomass production. In fed-batch mode, nutrients are periodically added during cultivation, extending the exponential growth phase and increasing biomass production. However, this method requires more nutrient monitoring and has more risk of contamination compared to batch operation. In semi-continuous mode, a portion of the culture is periodically harvested at a certain time point, and fresh medium is added to maintain growth. This method maintains high biomass productivity but needs periodic intervention, which increases the labor intensity and the risk of contamination. In continuous mode, part of the culture is continuously harvested while new medium is added to the system. This method offers stable biomass productivity and requires low labor cost, but it is more difficult to control and more prone to contamination compared to the other operation modes.

Carotenoids produced by microalgae can be categorized into primary and secondary types, each serving distinct functions. Primary carotenoids, such as β-carotene, lutein, fucoxanthin, lycopene, and violaxanthin, are essential for photosynthesis, playing a crucial role in light harvesting and protecting chlorophyll from photodamage. Secondary carotenoids, such as astaxanthin and canthaxanthin, are not directly involved in photosynthesis but can act as antioxidants, protecting the cells from damage caused by stressors such as high light intensity, nutrient deprivation, high salinity, or oxidative stress [17]. In response to these stress factors, secondary carotenoids can be overproduced in microalgae. Several primary carotenoids, such as β-carotene, also accumulate under stress conditions and act as secondary metabolites [18]. Despite the ability to cultivate microalgae for its high-value carotenoids with health benefits, only astaxanthin and β-carotene are produced on a commercial scale by *Haematococcus pluvialis* [19,20] and *Dunaliella salina* [21], respectively.

In contrast to astaxanthin and β-carotene, lutein production from microalgae is not yet commercially established. Currently, natural lutein is primarily produced from marigold flowers [5]. However, several microalgal species offer promising prospects for the production of this carotenoid. There is high interest in natural lutein because of its potential health benefits (Table 1). However, high culture costs and low carotenoid yields are bottlenecks for the commercialization of lutein from microalgae. Lin et al. (2015) estimated that microalgae need to have a lutein content exceeding 1% of their dry weight (DW) to be considered commercially viable for production [22]. Unlike secondary carotenoids, which can be overproduced under stress conditions, the overproduction of primary carotenoids like lutein is much more challenging due to its connection to the growth performance of microalgae. Cultivation conditions can be optimized to enhance the biomass productivity (g DW/L/day) and lutein content (mg/g DW) in microalgae. However, the optimal parameters for maximizing the lutein content can reduce the biomass productivity, thereby lowering the overall lutein productivity (mg/L/day) [23]. Therefore, a balanced approach is necessary to optimize both lutein content and biomass productivity for maximum overall lutein production. Metabolic engineering, which enables the targeted upregulation of specific carotenoid biosynthesis pathways, might offer a solution to enhance lutein productivity [5]. This review exclusively focuses on enhancing lutein content and productivity through strain improvement via random mutagenesis and metabolic engineering, alongside cultivation strategies such as fed-batch, batch, and semi-continuous methods. Unlike reviews that emphasize cultivation parameters for increasing carotenoid content broadly, our review specifically concentrates on lutein and offers a comparative analysis across various microalgal species.

Downstream processing (harvesting, cell disruption, extraction, and purification) of microalgal biomass for lutein production are also crucial steps determining the production costs but will not be covered here, as they were recently extensively reviewed by Gong and Bassi (2016) [6] and Zohra et al. (2022) [24].

## 3. Random Mutagenesis to Increase the Lutein Production

Random mutagenesis is a versatile and powerful technique employed in genetic research and biotechnology to induce genetic variation and enhance desirable traits in organisms [25]. This approach involves the deliberate induction of mutations across an organism’s genome without targeting specific genes, thus generating a diverse pool of genetic variants [26]. The method stands in contrast to new genomic techniques (NGTs), which involve specific, intentional changes to particular genes. The process of random mutagenesis typically involves exposing microalgal cultures to physical or chemical mutagens [25,26]. Physical mutagens such as UV radiation or gamma rays create DNA damage that can lead to mutations during DNA repair processes [25]. Chemical mutagens, like ethyl methanesulfonate (EMS) and *N*-methyl-*N′*-nitro-*N*-nitrosoguanidine (MNNG), introduce changes by chemically altering nucleotides, causing mispairing and subsequent mutations during replication [27]. Random mutagenesis does not require any previous knowledge about the genetics and metabolism of the target organism and the development of molecular tools, which can be time-consuming and expensive [28]. This advantage is particularly significant for organisms with limited or inaccessible genomic information and can be used for the assignment of phenotypes (traits/characteristics) to a certain gene/genotype and the expansion of the current understanding of the biology and metabolism of several microalgae species [26].

One of the significant applications of random mutagenesis is in the field of microalgal biotechnology, particularly for the enhancement of valuable compounds like lutein (Table 2). Mixotrophic cultivation of a *Chlorella sorokiniana* MB-1-M12 mutant, created by a 1 h MNNG treatment, resulted in a higher lutein content and lutein productivity (7.52 mg/g DW; 3.63 mg/L/day) compared to the wild-type strain *C. sorokiniana* MB-1 (5.86 mg/g DW; 2.56 mg/L/day) [29]. Ren et al. (2022) also treated microalga *Chromochloris zofingiensis* with MNNG, resulting in the mutant *Cz-pkg* that could reach a lutein content of 6.28 mg/g DW with a lutein productivity of 10.57 mg/L/day, which was 2.5- and 8.5-fold higher, respectively, than that of the wild-type [30]. Characterization of the mutant *Cz-pkg* with single-nucleotide polymorphism (SNP) analysis to pinpoint the exact mutation(s) revealed a T to A substitution in the cGMP-dependent protein kinase (PGK), leading to a premature stop codon (UAG) [30]. As a result, the photosystem of the *Cz-pgk* mutant can work effectively for uptake of both inorganic (CO_2_) and organic carbon sources (glucose) under mixotrophic conditions, thereby enhancing its capacity for lutein production [30].

Following mutagenesis, the challenge lies in screening and identifying mutants with the desired trait—in this case, increased lutein content. Random mutagenesis introduces mutations randomly, leading to many neutral or deleterious changes that necessitate extensive screening to identify beneficial mutants. This screening process can be labor-intensive, time-consuming, and resource-intensive [26]. Additionally, the randomness of the mutations can cause unintended effects, with changes occurring in genes unrelated to the targeted trait, resulting in unforeseen alterations in other phenotypic characteristics. Effective screening methods for lutein include the use of norflurazon and nicotine. Norflurazon, an herbicide that inhibits phytoene desaturase in the carotenoid biosynthesis pathway, disrupts the production of downstream carotenoids, including lutein [31]. Nicotine, a specific inhibitor of lycopene β-cyclase, affects lutein biosynthesis by interfering with the conversion of lycopene to β-carotene [32]. Screening for mutants that tolerate and thrive in the presence of norflurazon and/or nicotine allows for the identification of strains with modifications in the carotenoid biosynthesis pathway, potentially leading to an increased lutein content. Cordero et al. (2011) used these screening methods to identify mutants with a higher lutein content [32]. *C. zofingiensis* was chemically mutagenized with MNNG and spread onto media supplemented with norflurazon or nicotine. From all the mutants, two mutants resistant to norflurazon showed a 53–55% increase in lutein content relative to the wild-type, reaching values of 7.0 mg/g DW. One mutant resistant to nicotine exhibited a 1.4-fold higher lutein content than that of the wild-type, reaching 6.4 mg/g DW [32]. 

A complementary approach to identifying mutants with higher lutein content involves a color-based colony screening approach. This technique leverages the visual differences in pigmentation that result from variations in carotenoid content among different mutants. For example, mutants with a higher lutein content will appear more intensively yellow. Huang et al. (2018) used this screening approach to select for a higher zeaxanthin, lutein, and β-carotene content in *C. zofingiensis* cells treated with the chemical mutagen MNNG and grown on media containing glucose [33]. The selected mutant, *CZ-bkt1*, with a premature stop codon in β-carotene ketolase, accumulated zeaxanthin up to 0.34 mg/g DW, lutein up to 7.12 mg/g DW, and β-carotene up to 1.51 mg/g DW [33]. Stress conditions known to induce carotenoid biosynthesis were used to further increase the carotenoid content in the mutant. High-light radiation (460 µmol/m^2^/s) and nitrogen deficiency could increase the zeaxanthin content up to 7.00 mg/g DW, lutein up to 13.81 mg/g DW, and β-carotene up to 7.18 mg/g DW when induced by high-light irradiation and nitrogen deficiency [33].

## 4. Metabolic Engineering for Enhanced Production of Lutein

In contrast to random mutagenesis, new genomic techniques (NGTs) can introduce very specific, intentional changes to particular genes. Metabolic engineering using NGTs is a powerful approach for steering cell metabolism by modifying specific pathway enzymes or regulatory proteins. It utilizes a toolbox that includes gene knock-out/knock-in techniques, as well as gene repression and overexpression applications. Gene knock-out techniques, such as clustered regulatory interspaced short palindromic repeats/Cas (CRISPR/Cas) [34], zinc-finger nucleases (ZFNs) [35], and transcription activator-like effector nucleases (TALENs) [36], result in the modification of specific genes to abolish undesired pathways [37]. Knock-in techniques integrate new genetic material to introduce novel functionalities. Gene repression restricts the expression of certain genes to redirect metabolic flux and includes RNA interference (RNAi), a technique used to degrade specific mRNA molecules, preventing their translation into proteins [38]. CRISPR interference (CRISPRi), another technique under gene repression, interferes with the transcription machinery [39]. Gene overexpression techniques, achieved through promotor engineering or copy number amplification, can generate a many-fold overexpression of a specific gene that is responsible for a rate-determining step. Metabolic engineering techniques are key approaches to improve microalgal biomass productivity and suitability on an industrial scale in terms of high-value compound production (such as lutein), growth rate, mode of nutrition, and synergism between lutein accumulation and growth [5].

Metabolic engineering relies on a good knowledge of the biosynthetic pathway and involved genes. The carotenoid biosynthetic pathway in microalgae is intricate and involves several enzymatic steps, each of which can serve as a potential target for metabolic engineering. Precursors of the carotenoid biosynthesis pathway can be retrieved from the mevalonate (MVA) pathway and/or the methyl-D-ertythritol phosphate pathway (MEP). However, in green microalgae, the MVA pathway has been lost, leaving the MEP pathway as the sole pathway for isopentenyl phyrophosphate (IPP) synthesis. The carotenoid biosynthetic pathway is well described in Figure 1 [39,40].

Regulatory mechanisms in this pathway include transcriptional regulation, where the genes encoding carotenoid biosynthetic enzymes are regulated by (1) environmental factors such as light and nutrient availability, (2) feedback inhibition, where end products inhibit upstream enzymes, and (3) post-transcriptional regulation, which influences mRNA stability and translation efficiency [43]. One important example of post-transcriptional regulation is by the Orange (Or) gene, which is known to affect carotenoid accumulation and is involved in stabilizing the PSY gene, ensuring that PSY remains active and functional within the cell [44]. Different strategies can be used to increase the content of carotenoids, either directly by overexpression of endogenous enzymes involved in the carotenoid biosynthesis pathway, or indirectly by inhibition of competition pathways and thus redistributing the metabolic flow towards certain pigments [5].

Applications of metabolic engineering in microalgae for overproduction of carotenoids, and lutein in particular, are listed in Table 2. To increase the overall carotenoid production, the PSY gene and Or gene are good candidates for metabolic engineering, as they catalyze a rate-limiting step in the carotenoid biosynthesis pathway. Overexpression of the Or gene in *Chlamydomonas reinhardtii* using a strong dual-promotor system led to a lutein production from 0.69 mg/L to 1.04 mg/L and a β-carotene production from 0.18 mg/L to 0.24 mg/L, respectively [45]. Functional analysis of Or genes in various orange-flesh melon fruits identified a single-nucleotide polymorphism known as “golden SNP”, which converts a highly conserved arginine to histidine and is responsible for an elevated carotenoid accumulation [44]. Site-directed mutagenesis was used by Yazdani et al. (2021) to introduce this “golden SNP” in the Or gene in *C. reinhardtii* and overexpression resulted in a 4-fold increase in α-carotene, 3.1-fold increase in lutein, 3.2-fold increase in β-carotene, and 3.1-fold increase in violaxanthin [46]. Overexpression of the PSY gene itself can also result in a higher overall carotenoid content, as shown by Velmurugan et al. (2023) [47]. The overexpression of the endogenous PSY gene in *D. salina* led to a substantial increase in lutein and β-carotene content, up to 5.4-fold and 7.2-fold compared to the wild-type [47].

Considering the metabolic branches originating from lycopene, overexpression of LCYB or LCYE could affect the productivity of carotenoids in each branch pathway. The overexpression of LCYE could redirect the lycopene flux towards the α-branch, with an increase in lutein production as a result. Tokunuaga et al. (2021) overexpressed the LCYE gene in *C. reinhardtii* using a dual-promotor system, resulting in a significant increase in lutein concentrations (2.3-, 2.0-, and 2.6-fold, respectively) compared to the wild-type [48]. Lou et al. (2021) used the LCYE gene from *C. vulgaris*, which is rich in lutein, for the heterologous expression in *C. reinhardtii*, where it showed an enhanced lutein content from 0.583 mg/g DW to 1.357 mg/g DW [49]. Both studies indicate that the conversion of lycopene to α-carotene can be increased by homologous or heterologous expression of the LCYE gene.

**Table 2 marinedrugs-22-00329-t002:** Application of random mutagenesis and metabolic engineering in microalgae to increase carotenoid content.

Source Species	Host Strain	Target Gene	Technique	Carotenoid	Reference
*C. sorokiniana*	/	/	Random mutagenesis (MNNG treatment)	Increased lutein content up to 7.25 mg/g DW with a productivity of 2.56 mg/L/day.	[29]
*C. zofingiensis*	/	/	Random mutagenesis (MNNG treatment)	Increased lutein content up to 6.25 mg/g DW with a productivity of 10.57 mg/L/day	[30]
*C. sorokiniana*	/	/	Random mutagenesis (MNNG treatment)	Increased lutein content up to 7.0 mg/g DW and 6.4 mg/L/day	[32]
*C. zofingiensis*	/	/	Random mutagenesis (MNNG treatment)	Increased zeaxanthin (up to 7.0 mg/g DW), lutein (up to 13.81 mg/g DW) and β-carotene (7.18 mg/g DW).	[33]
*C. reinhardtii*	Endogenous	Or	Overexpression Or gene using dual-promotor system	Lutein production increase from 0.69 mg/L to 1.04 mg/L and from 0.18 mg/L to 0.24 mg/L	[45]
*C. reinhardtiii*	Endogenous	Or	Overexpression Or gene	Increased α-carotene (1.9-fold higher), lutein (2-fold higher), β-carotene (2.1-fold higher) and violaxanthin (2.1-fold higher) content compared to WT.	[46]
*C. reinhardtii*	Endogenous	Or	Overexpression Or gene with single amino acid substitution using site-directed mutagenesis	Increased α-carotene (4-fold higher), lutein (3.1-fold higher), β-carotene (3.2-fold higher) and violaxanthin (3.1-fold higher) content compared to WT.	[46]
*C. reinhardtii*	*Brassica oleracea*	Or	Heterologous expression Or gene.	Increased lutein (1.5-fold higher: 112.4 pg/cell to 73.0 pg/cell lutein (WT)) and astaxanthin content (2-fold higher: 0.41 pg/cell to 0.2 pg/cell (WT))	[50]
C. *reinhardtii*	*Mesorhizobium loti* and *Sulfurihydrogenibium yellowstonense*	CA	Heterologous expression of CA gene	Increased lutein concentration from 4.41 mg/L (WT) to 8.89 mg/L (CA from *Ml)* and 7.07 mg/L (CA from *SY*).	[51]
*D. salina*	Endogenous	PSY	Overexpression of PSY gene.	Increased lutein (7.6-fold higher) and β-carotene (5.4-fold higher) content compared to WT.	[47]
*D. salina*	*H. pluvialis*	PSY	Heterologous expression of PSY gene.	Increased lutein (7.2-fold higher) and β-carotene (2.4-fold higher) conten compared to WT.	[47]
*C. reinhardtii*	*D. salina*	PSY	Heterologous expression LCYE gene.	Increased lutein (2.6-fold higher) content compared to WT.	[32]
*Scenedesmus*	*/*	PSY	Expression of synthetic PSY gene.	Increased β-carotene content from 10.8 mg/g (WT) cell to 30 mg/g cell.	[28]
*C. reinhardtii*	Endogenous	LCYE	Overexpression of LCYE gene.	Increased lutein (at least 2-fold higher) content.	[48]
*C. reinhardtii*	*C. vulgaris*	LCYE	Heterologous of LCYE gene.	Increased lutein content (2.3-fold higher) compared to WT.	[49]
*H. pluvialis*	Endogenous	PDS	Overexpression of PDS gene with single amino acid substitution using site-directed mutagenesis.	Increased lutein (1.5 µg/g DW to 1.9 µg/g DW), zeaxanthin (142 µg/g DW to 214 µg/g DW), β-carotene (532 µg/g DW to 728 µg/g DW) and astaxanthin content compared to WT.	[52]
*C. zofingiensis*	Endogenous	PDS	Overexpression of PDS gene with single amino acid substitution using site-directed mutagenesis.	Increased total carotenoid content with 32.1% and astaxanthin with 54.1%.	[53]
*C. reinhardtii*	/	LCYE	Knock-out of the LCYE gene using CRISPR/Cas (NHEJ)	Increased zeaxantin content (up to 60%).	[54]
*C. reinhardtii*	/	LCYE	Knock-out of the LCYE gene using CRISPR/Cas (HDR)	Increased zeaxantin (0.31 mg/L (WT) to 0.59 mg/L), antheraxanthin (0.28 mg/L (WT) to 0.63 mg/L) and violaxanthin (1.3 mg/L (WT) to 2.3 mg/L) content.	[55]

According to the studies mentioned, Or, PSY, and ε-LCY are the key targets for metabolic engineering to increase the lutein content. PSY regulates the overall flow towards carotenoid synthesis, while ε-LCY directs the flow specifically towards the α-carotene branch [56]. Therefore, overexpression of the PSY gene influences the total carotenoid content, while overexpression of the ε-LCY gene only positively affects the lutein content. Aside from the genes involved in the carotenoid biosynthetic pathway, other genes involved in microalgae metabolism might be good targets for metabolic engineering. Lin et al. (2022) showed that the heterologous expression of carbonic anhydrases (CAs) in *C. reinhardtii* with the aim to increase photosynthetic capability and carbon capture could increase biomass from lutein production from 4.41 mg/L to 8.89 and 7.07 mg/L [51].

## 5. Cultivation Strategies to Increase Microalgal Lutein Production

Many microalgal species, including *Chlorella* sp., *Chlamydomonas* sp., *Desmosdesmus* sp., and *Scenedesmus* sp., produce lutein [57]. An important step before implementing a cultivation strategy is understanding the effect of cultivation parameters on lutein production of microalgae, which is reviewed by Hu et al. (2018) [16], Liu et al. (2021) [8], and Suparmaniam et al. (2024) [23]. Under photoautotrophic or mixotrophic conditions, high light intensity (often > 750 µmol/m^2^/s) induces carotenogenesis, leading to higher lutein content in microalgae [23]. Furthermore, adequate CO_2_ supply and the removal of the built-up O2 are essential for biomass growth, especially in photoautotrophic conditions. Under heterotrophic conditions, glucose is the optimal carbon source, and urea is the most effective nitrogen source for lutein production, outperforming carbon sources, such as glycerol and acetate, and nitrogen sources, such as nitrate and ammonium [58,59,60,61]. Under mixotrophic cultivation, acetate is the preferred carbon source, while nitrate and urea are the preferred nitrogen sources for lutein production. Importantly, high initial concentrations of carbon sources (often > 50–100 g/L) such as glucose and acetate can inhibit growth and reduce lutein production due to substrate inhibition [16]. Aeration also strongly affects lutein productivity since the heterotrophic metabolism of microalgae requires oxygen for growth [60]. Other than supplying sufficient oxygen for biomass production, oxygen levels do not influence the lutein content [16]. Nitrogen availability is critical for biomass productivity and lutein production, as nitrogen depletion leads to lutein degradation [57]. For certain microalgal species such as *Chlamydomonas* and *Scenedesmus* species, it has been reported that the lutein content remains high until the onset of nitrogen depletion [61,62]. Finally, each microalga has a specific optimal pH and temperature range for growth. Generally, increasing temperature up to a stress limit enhances lutein productivity, while temperatures beyond this threshold severely affect cellular growth and survival. On the contrary, cooler temperatures decrease the nutrient uptake rate and slow down both microalgal growth and lutein productivity [57]. However, Léon-Vaz et al. (2023) demonstrated that certain microalgal species produce higher lutein content at elevated light intensity (500 µmol/m^2^/s) and 10 °C, compared to lower light intensity (100 µmol/m^2^/s) and 20 °C [63]. Consequently, the optimal temperature for lutein production varies by species and must be determined experimentally.

In the following part of this review, innovative cultivation strategies for various microalgae species will be discussed, focusing on lutein productivity (mg/L/day) and lutein content (mg/g DW). The strategies resulting in the highest lutein production are shown in Table 3 and Table 4.

### 5.1. Cultivation Strategies for Chlorella Species

Mixotrophic and heterotrophic conditions have been explored for lutein production in various *Chlorella* species, including *C. sorokiniana, C. protothecoides* (now *Auxenochlorella protothecoides*), and *C. vulgaris*. Researchers place particular emphasis on cultivation strategies to maximize lutein production in strains of *C. sorokiniana*. *C. sorokiniana* FZU60 is an excellent candidate for large-scale lutein production due to its rapid growth and high lutein content, reaching 8.29–11.22 mg/g DW under photoautotrophic and mixotrophic conditions, and 2.33–4.42 mg/g DW under heterotrophic conditions. Based on the reviewed articles (Table 3), the optimal conditions for cultivating *C. sorokiniana* FZU60 include a temperature range of 30–33 °C, light intensity of 150–750 μmol/m^2^/s depending on culture density, and an optimal pH of 7.5. Suitable media are BG-11 and Mann and Meyer’s. Depending on the culture density, the aeration requirements vary around 0.02–0.2 vvm with 2–2.5% CO_2_ for photoautotrophic or mixotrophic growth. For optimal heterotrophic growth, the dissolved oxygen (DO) is crucial and should be controlled at 20–50% air saturation using aeration or agitation (stirring).

Two-stage strategies, starting from fed-batch mixotrophic conditions and transitioning to photoautotrophic conditions, achieve higher lutein productivities for *C. sorokiniana* FZU60 compared to single-stage batch and fed-batch modes under mixotrophic conditions. Xie et al. (2020) used a two-stage strategy to maximize lutein productivity of *C. sorokiniana* FZU60 [64]. Initially, this strain was cultivated under fed-batch mixotrophic conditions for 3 days under white light 350 μmol/m^2^/s, maintaining 1 g/L sodium acetate (NaAc) by adding a 400 g/L NaAc solution when the DO reached 7 mg/L. The culture was then shifted to photoautotrophic conditions for 4 days using the same light conditions, with concentrated BG11 medium added when 90% of the nitrate was consumed. This strategy achieved a higher lutein productivity (4.67 mg/L/day) compared to a batch mixotrophic operation (3.59 mg/L/day), while achieving a lutein content of 9.51 mg/g. Furthermore, Xie et al. (2019) discovered that a two-stage strategy to produce lutein in *C. sorokiniana* FZU60 can be operated in semi-continuous mode [65]. Initially, *C. sorokiniana* FZU60 was cultivated under fed-batch mixotrophic conditions in BG-11 medium with 1 g/L NaAc pulses every 12 h for 1.5 days under white light (150 μmol/m^2^/s). Subsequently, 92.5% of the culture was shifted to photoautotrophic conditions with the same light intensity and 100-fold BG11 medium for another 1.5 days. The other 7.5% was used as the seed for a new cycle in the first stage. This strategy enhanced the average lutein productivity to 11.57 mg/L/day, compared to 4.78 mg/L/day in batch culture and 4.20 mg/L/day in fed-batch culture, while reaching a lutein content of on average 9.57 mg/g DW after each cycle.

The nutrient feeding dosage and light intensity also significantly impact the lutein productivity. Ma et al. (2020) compared different fed-batch strategies for *C. sorokiniana* FZU60 in BG-11 medium under mixotrophic conditions (750 μmol/m^2^/s) [66]. They found that feeding a constant NaAc concentration of 1 g/L yielded higher lutein productivity (8.04 mg/L/day) compared to the gradient feeding of NaAc (6.11 mg/L/day). Additionally, the authors explored the effect of the light intensity of the second stage. Higher lutein productivity and lutein content were achieved with 750 μmol/m^2^/s (8.25 mg/L/day; 11.22 mg/g DW) compared to 150 μmol/m^2^/s (5.62 mg/L/day; 9.85 mg/g DW).

Xie et al. (2022) investigated different nutrient feeding strategies for *C. sorokiniana* FZU60 under heterotrophic conditions in a 5 L fermenter [59]. Importantly, the DO was controlled at 20% by adjusting the stirring speed. When DO levels increased, new media with glucose were fed into the fermenter. Enhancements in lutein productivity and lutein concentration in the fermenter were observed using a 3-fold concentration of Mann and Myer’s medium with urea (82.50 mg/L/day; 415.93 mg/L after 6 days) compared to 1-fold (53.93 mg/L/day; 273.86 mg/L after 6 days) and 6-fold (50.37 mg/L/day; 254.90 mg/L after 6 days) concentrations. This observation was explained by the more stable substrate addition and operational parameters (stirring speed and DO) compared to the 1-fold, and less substrate inhibition compared to the 6-fold. Notably, the lutein content of *C. sorokiniana* FZU60 under fed-batch heterotrophic growth was only 2.57 mg/g DW.

**Table 3 marinedrugs-22-00329-t003:** Cultivation strategies to increase the lutein production in *Chlorella* sp.

Microalgae	Cultivation Mode	Reactor Volume (L)	Strategy *	Lutein Content (mg/g DW)	Lutein Production (mg/L)	Lutein Productivity (mg/L/day)	Reference
*C. sorokiniana* FZU60	Two-stage semi-continuous (5 cycles)	1	1st Fed-batch Mixo (BG-11 with 1 g/L NaAc every 12 h and 150 μmol/m^2^/s) After 1.5 days, 92.5% medium replacement which is transferred to 2nd stage. 7.5% to new cycle 1st stage. 2nd Batch Photo (150 μmol/m^2^/s)	9.57 (Day 3, average of the 5 cycles)	17.35 (Day 3, average of the 5 cycles)	11.57 (average of the 5 cycles)	[65]
*C. sorokiniana* FZU60	Two-stage	50	1st Fed-batch Mixo (acetate and 350 μmol/m^2^/s) 2nd Fed-batch Photo (BG11 and 350 μmol/m^2^/s)	9.51 (Day 7)	33.55 (Day 7)	4.67 (average over 7 days)	[64]
*C. sorokiniana* FZU60	Fed-batch	1	Mixo (acetate and 750 μmol/m^2^/s)	8.29 (Day 7)	32.16 (Day 4)	8.04 (average over 4 days)	[66]
*C. sorokiniana* FZU60	Two-stage	1	1st Fed-batch Mixo (acetate and 750 μmol/m^2^/s) 2nd Batch Photo (750 μmol/m^2^/s)	11.22 (Day 8)	65.96 (Day 8)	8.25 (average over 8 days)	[66]
*C. sorokiniana* FZU60	Fed-batch	5	Hetero (Mann and Myer’s with glucose and urea)	2.57 (Day 6)	415.93 (Day 6)	82.50 (average)	[59]
*C. sorokiniana* MB-1-M12	Semi-continuous	1	Batch Mixo (acetate and 150 μmol/m^2^/s) After glucose depletion, 75% medium replacement	4.98 (Day 7 in 2nd cycle)	11.95 (Day 7 in 2nd cycle)	6.61 (2nd cycle average)	[67]
*C. sorokiniana* MB-1-M12	Batch	1	Batch Photo (150 μmol/m^2^/s)	6.01 (Day 4)	16.40 (Day 5)	3.56 (average)	[68]
*C. sorokiniana* MB-1-M12	Batch	1	Batch Mixo (acetate and 150 μmol/m^2^/s)	7.00 (Day 5)	18.04 (Day 5)	5.15 (average)	[68]
*C. sorokiniana* MB-1-M12	Batch	1	Batch Hetero (glucose)	2.31 (Day 7)	7.71 (Day 4)	1.88 (average)	[68]
*C. sorokiniana* MB-1-M12	Two-stage	1	1st Batch Photo (150 μmol/m^2^/s) 2nd Batch Hetero (glucose)	4.75 (Day 10)	24.97 (Day 10) 20.5 (after day 6)	1.75 (average)	[68]
*C. sorokiniana* MB-1-M12	Two-stage	1	1st Batch Hetero (glucose) 2nd Batch Photo (150 μmol/m^2^/s)	6.52 (Day 6)	34.62 (Day 9)	2.86 (average)	[68]
*C. sorokiniana* MB-1-M12	Two-stage	1	1st Batch Mixo (acetate and 150 μmol/m^2^/s) 2nd Batch Hetero (glucose)	3.50 (Day 10)	19.07 (Day 10) 17.5 (after day 7)	1.3 (average)	[68]
*C. sorokiniana* MB-1-M12	Two-stage	1	1st Batch Hetero (glucose) 2nd Batch Mixo (acetate and 150 μmol/m^2^/s)	6.17 (Day 10)	33.64 (Day 10)	3.42 (average)	[68]
*C. sorokiniana* MB-1-M12	Fed-batch	1	Fed-batch Hetero (glucose)	3.40 (Day 11)	39.50 (Day 11)	3.24 (average)	[69]
*C. sorokiniana* MB-1-M12	Two-stage semi-continuous (3 cycles)	1	1st Fed-batch Hetero (glucose and urea) After highest biomass accumulation, 75% medium replacement which is transferred to 2nd stage. 25% to new cycle 1st stage. 2nd Batch Mixo (acetate and 150 μmol/m^2^/s)	6.77 (1ste cycle; day 11) 6.61 (2nd cycle, day 17) 6.53 (3th cycle, day 23)	76.00 (1ste cycle; day 11) 80.88 (2nd cycle, day 17) 81.77 (3th cycle, day 23)	1st stage (±6.17 average) 2nd stage (±2.86 average)	[69]
*C. sorokiniana* MB-1-M12	Two-stage semi-continuous (3 cycles)	5	1st Fed-batch Hetero (glucose and urea) After highest biomass accumulation, 75% medium replacement which is transferred to 2nd stage. 25% to new cycle 1st stage. 2nd Batch Mixo (acetate and 150 μmol/m^2^/s)	8.19 (1ste cycle; day 14) 8.09 (2nd cycle, day 18) 8.71 (3th cycle, day 21)	181.11 (1ste cycle; day 14) 153.60 (2nd cycle, day 18) 169.17 (3th cycle, day 21)	1st stage (±20.02 average) 2nd stage (±5.55 average)	[69]
*Chlorella protothecoides* CS-41	Two-stage	30	1st Fed-batch Hetero (glucose and urea) After 10 days temperature shifted from 28 °C to 32 °C 2nd Batch (nutrient limited phase)	5.35 (Day 14) 3.8 (after 10 days)	209.08 (Day 14) 200 (after 10 days)	19.18 (average)	[70]
*Chlorella minutissima* MCC-27	Batch	2	Batch Photo (Constant 260 µmol/m s)	6.37 (Day 5)	22.1 (Day 5)	4.32 (average)	[71]
*Chlorella minutissima* MCC-27	Batch	2	Batch Photo (linear increase from 75 µmol/m s to 260 µmol/m s)	8.24 (Day 5)	26.75 (Day 5)	5.35 (average)	[71]
*Chlorella vulgaris*	Fed-batch	5	Hetero (glucose and urea)	5.32 (Day 5)	252.75 (Day 5)	67.4 (average)	[72]

* Photoautotrophic growth (photo), mixotrophic growth (mixo), and heterotrophic growth (hetero).

*C. sorokiniana* MB-1-M12 is another promising strain for lutein production, developed through random mutagenesis with MNNG [31]. This lutein-rich mutant has lutein content ranging from 4.98 to 8.71 mg/g DW under phototrophic and mixotrophic conditions, and from 2.31 to 4.9 mg/g DW under heterotrophic conditions. While optimal parameters for pH, light intensity, and aeration are similar to previously described *C. sorokiniana* FZU60, the optimal temperature range for maximizing lutein production in *C. sorokiniana* MB-1-M12 is slightly lower, at 27–28 °C. Chen et al. (2019) showed that the medium replacement ratios used in semi-continuous mode also impact the lutein productivity of *C. sorokiniana* MB-1-M12 under mixotrophic conditions (150 µmol/m^2^/s with NaAc). They evaluated three medium replacement ratios (25%, 50%, 75%) over six repeated cycles, replacing the medium when carbon was depleted [67]. Semi-continuous cultivation with 75% medium replacement (6.61 mg/L/day) resulted in higher lutein productivity than batch (3.43 mg/L/day) and other replacement ratios (50%: 3.79 mg/L/day; 25%: 2.76 mg/L/day).

Chen et al. (2021) evaluated four different two-stage strategies for *C. sorokiniana* MB-1-M12 in BG-11 medium over 10 days: TSAH—initiated as autotrophic and switched to heterotrophic on day 3, TSHA—initiated as heterotrophic and switched to autotrophic on day 4, TSMH—initiated as mixotrophic and switched to heterotrophic on day 5, and TSHM—initiated as heterotrophic and switched to mixotrophic on day 4 [68]. The strategies that started with heterotrophic conditions obtained higher maximum lutein concentration and lutein content compared to the strategies initiated with photoautotrophic and mixotrophic conditions and ending with heterotrophic conditions (Table 3). In addition, Chen et al. (2022) further improved the TSHM strategy by integrating fed-batch and semi-continuous operational strategies for the *C. sorokiniana* MB-1-M12 [69]. During the heterotrophic phase, glucose and urea were fed into the reactor to maintain glucose concentrations between 2.0 and 7.5 g/L and ensure sufficient nitrogen. When the highest biomass was reached in the reactor, the fed-batch heterotrophic mode was switched to mixotrophic by transferring 75% of the medium to the mixotrophic mode, while the remaining 25% continued in a new fed-batch heterotrophic phase. This strategy was applied for three cycles in a 1 L reactor, achieving lutein production (average of 79.55 mg/L) and lutein content (average of 6.44 mg/g DW). In addition, this strategy was successfully scaled to a 5 L reactor, achieving lutein production (average of 167.96 mg/L) and lutein content (average of 8.33 mg/g DW).

Finally, Shi et al. (2002) investigated a strategy to enhance lutein productivity in *Chlorella protothecoides* CS-41 (*Auxenochlorella protothecoides*) under heterotrophic conditions in a 30 L fermenter [70]. The strategy involved initiating a fed-batch culture with glucose (40 g/L) and urea (7 g/L) for 10 days at 28 °C. Subsequently, the culture was exposed to nutrient-limited conditions at 32 °C for 84 h (equivalent to 3.5 days). After 10 days, the lutein concentration in the reactor reached 200 mg/L, with the strain exhibiting a lutein content of 3.8 mg/g DW. Following the period of nutrient limitation of 3.5 days, the lutein content increased to 5.35 mg/g DW. However, the lutein concentration only slightly rose to a maximum of 209.08 mg/L due to a decline in microalgal biomass in the reactor.

### 5.2. Cultivation Strategies for Other Microalgal Species

Several species of *Scenedesmus* exhibit the capacity to produce lutein. These green algae thrive in freshwater habitats and can resist high light intensities (1625–1900 μmol/m^2^/s). The optimal lutein production occurs at temperatures of 30–40 °C and the preferred pH range for cultivation is between 6 and 8, with Mann and Myers medium commonly employed for lutein production. Aeration levels of 0.2–0.5 vvm, coupled with CO_2_ supplementation ranging from 2.5 to 10%, enhance growth. Under photoautotrophic conditions, lutein content is 4.8–5.5 mg/g DW, around 2.55 mg/g DW under mixotrophic conditions, and around 1.49 mg/g DW under heterotrophic conditions, which is generally lower compared to *Chlorella* species (Table 3 and Table 4). Ho et al. (2014) selected *S. obliquus* FSP-3 from the six *S. obliquus* strains in photoautotrophic conditions for its higher lutein productivity capacities [61]. Different light-related strategies were evaluated for this strain in batch, including various types of fluorescent lamps (e.g., TL5, T8, and helix lamps) and light intensities (from 30 to 600 µmol/m^2^/s). The full white light spectrum (410–610 nm) is more favorable for lutein production than monochromatic green (480–580 nm), blue (435–515 nm), and red (600–690 nm) LED light sources. The optimal lutein productivity (4.08 mg/L/day) was obtained when using a white TL5 fluorescent lamp at a light intensity of 300 µmol/m^2^/s. Florez-Miranda et al. (2017) tested a two-stage strategy to increase the lutein productivity of *S. incrassatulus* CLHE-Si01 [73]. The heterotrophic stage was performed in batch mode using glucose as a carbon source. Once glucose was consumed, the cultures were transferred to a photoautotrophic stage (230 µm/m^2^/s). After 24 h of photoinduction, the lutein productivity reached 3.10 mg/L/day. Chen et al. (2019) found that a fed-batch strategy under mixotrophic conditions (150 μmol/m^2^/s, 12 h/12 h) continuously feeding Detmer’s medium with 20 g/L glucose led to a lutein productivity of 4.96 mg/L/day for *S. obliquus* CWL-1 [74].

The freshwater microalga *Desmodesmus* also has the capacity to accumulate lutein in its cells. Xie et al. (2013) identified *Desmodesmus* sp. F51 as the best strain for lutein production [75]. Various growing media were tested, and Modified Bristol’s medium was selected as optimal for lutein productivity (3.05 mg/L/day) under phototrophic conditions (150 µmol/m^2^/s). To further enhance the lutein production, a fed-batch strategy was employed using different nitrate concentrations (1.1, 2.2, 4.4, 8.8, and 17.6 mM). The highest lutein productivity (3.56 mg/L/day) and content (5.05 mg/g DW) were achieved with 2.2 mM nitrate pulse-feeding, with minimal differences observed between other concentrations. Interestingly, Ahmed et al. (2019) discovered that the synergistic effect of the plant hormones salicylic acid and succinic acid can enhance nitrate assimilation and increase lutein production in *Desmodesmus* sp. [76]. Supplementing with 100 μM salicylic acid and 2.5 mM succinic acid under phototrophic conditions in batch culture achieved a maximal lutein content of 7.01 mg/g DW and lutein productivity of 5.11 mg/L/day. Additionally, a fed-batch strategy involving nitrate, succinic acid, and salicylic acid further enhanced the lutein content and lutein productivity, reaching 7.50 mg/g DW and 5.78 mg/L/day, respectively.

*Chlamydomonas* is a green microalga with high light resistance and the capacity to produce lutein in fresh and salt water. Similar to *Scenedesmus* and *Desmodesmus*, fewer cultivation strategies have been explored to increase lutein production in this genus. Different light and temperature strategies were evaluated for *Chlamydomonas* sp. [77]. However, the reported lutein productivities and lutein content (2.52–4.24 mg/g DW) are generally lower compared to those of *Chlorella* (Table 3). However, as discussed earlier, its well-understood genetics make it an ideal candidate for genetic manipulation, allowing us to potentially enhance its lutein production.

Based on the literature reviewed, *Chlorella* sp. appears to be superior to *Scenedesmus* sp., *Chlamydomonas* sp., and *Desmodesmus* sp. for lutein production. The most effective cultivation strategy, yielding the highest lutein productivity (82.50 mg/L/day), involves cultivating *C. sorokiniana* FZU60 under heterotrophic conditions in fed-batch mode with 3-fold concentrated Mann and Myer’s medium supplemented with glucose and urea, while controlling dissolved oxygen (DO) in the fermenter [59]. Over 6 days, this approach achieves a lutein concentration of 415.93 mg/L due to high biomass concentration in the fermenter. Controlling operational parameters such as pH, ensuring effective gas supply (ambient air and CO_2_), and maintaining dissolved oxygen (DO) levels in the fermenter are crucial for achieving higher biomass production and consequently enhancing lutein productivity. However, under these heterotrophic conditions, the lutein content is only 2.57 mg/g DW. To enhance the lutein content in *Chlorella*, two-stage strategies can be employed. For instance, using fed-batch heterotrophic conditions followed by mixotrophic conditions for *C. sorokiniana* MB-1-M12, as proposed by Chen et al. (2022) [69], results in a lutein content of 8.19 mg/g DW after 14 days of cultivation. Conversely, the fed-batch mixotrophy followed by mixotrophic conditions for *C. sorokiniana* FZU60, as suggested by Ma et al. (2020) [66], achieves a lutein content of 11.22 mg/g DW after 8 days of cultivation. These two-stage strategies were demonstrated to run in semi-continuous mode, optimally with a medium replacement ratio of 75% or higher.

**Table 4 marinedrugs-22-00329-t004:** Cultivation strategies to increase the lutein production in other microalgae.

Microalgae	Cultivation Mode	Reactor Volume (L)	Strategy *	Lutein Content (mg/g DW)	Lutein Production (mg/L)	Lutein Productivity (mg/L/day)	Reference
*Scenedesmus almeriensis*	Batch	2	Photo (1625 μE/m^2^/s)	5.5	/	4.77	[78]
*Scenedesmus almeriensis*	continuous mode (dilution rate 0.3 L/day)	2	Photo (1625 μE/m^2^/s)	5.4	/	3.8	[79]
*Scenedesmus obliquus* FSP-3	Batch	1	Photo (white TL5 fluorescent 300 µmol/m^2^/s)	4.80 (Day 5)	20.5 (Day 5)	4.08 (average)	[61]
*Scenedesmus incrassatulus* CLHE-Si01	Two-stage	6	1st Batch Hetero (glucose) After glucose was consumed 2nd Batch Photo (150 μmol/m^2^/s)	1.49 (Day 7)	/	3.10 (average)	[73]
*Scenedesmus obliquus* CWL-1	Fed-batch	7	Mixo (glucose and 150 μmol/m^2^/s 12 h/12 h)	2.55 (Day 9)	27.3 (Day 9)	4.96 (Day 5)	[74]
*Chlamydomonas* sp. JSC4	Batch	1	Photo (625 μmol/m^2^/s)	3.82	/	5.08	[62]
*Desmodesmus* sp. F51	Fed-batch	1	Photo (nitrate and 150 µmol/m^2^/s)	5.05 (Day 6)	16.5 (Day 6)	3.56 (Day 6)	[75]
*Desmodesmus* sp.	Fed-batch	1	Photo (nitrate, succinic acid and salicylic acid)	7.5 (Day 4)	18.9 (Day 6)	5.78	[76]
*Coccomyxa onubensis*	Batch		Photo (100 mM NaCl)	6.7 (Day 3)		1.63	[80]

* Photoautotrophic growth (photo), mixotrophic growth (mixo), and heterotrophic growth (hetero).

## 6. Future Directions and Challenges

### 6.1. Conclusion and Future Directions in Metabolic Engineering for Microalgal Lutein Production

Random mutagenesis and metabolic engineering of microalgae for enhanced lutein production presents a promising avenue for sustainable and efficient lutein biosynthesis, as both techniques were able to increase the lutein content in microalgae (Table 2). On the one hand, random mutagenesis involves the alteration of genetic material in a non-specific manner, leading to the discovery of beneficial mutations that can increase lutein production. On the other hand, metabolic engineering allows for the precise modification of specific genes known to be involved in the lutein biosynthetic pathway, offering a targeted approach to enhancing production.

Comparing studies to determine the most efficient different random mutagenesis and metabolic engineering technique is challenging because lutein content is often reported in different units, such as mg/g DW, pg/cell, or not reported at all. Overall, random mutagenesis resulted in a higher lutein content compared to metabolic engineering (Table 2), largely because it has been applied to microalgal species that already exhibit a high natural lutein production. On the other hand, metabolic engineering has predominantly focused on *C. reinhardtii*, primarily because its genome is one of the first to be fully sequenced, and it has well-established transformation protocols.

In this review, the highest found lutein content (13.81 mg/g DW) was from *C. zofingiensis* mutant, obtained via random mutagenesis, exposed to stress conditions (nitrogen deficiency and high light irradiation of 460 µmol/m^2^/s) [33]. However, this was achieved at a very small scale (50–100 mL), and further evaluation of lutein productivity is necessary to compare it with reported values from cultivation strategies (Table 3 and Table 4). Exploring various cultivation strategies could be beneficial and intriguing avenues for further research, as demonstrated by Chen et al. in 2017 with their *C. sorokiniana* MB-1-M12 mutant generated through random mutagenesis [29].

Interestingly, some genes modified through random mutagenesis, such as the cGMP-dependent protein kinase, have shown significant effects on lutein biosynthesis [30]. The identification of such genes is valuable because it highlights specific genetic targets that can be further studied and precisely modified through metabolic engineering. With the increasing availability of genomic knowledge and techniques, metabolic engineering research can shift to microalgal species, such as *Chlorella* sp. and *Scenedesmus* sp., that naturally produce higher levels of lutein.

In the context of utilizing microalgae biomass with enhanced lutein content, consideration must be given to two key regulatory aspects: novel food regulation and genetically modified organism (GMO) legislation. Some dried microalgal biomass with high lutein content can be consumed directly or after cell disruption to enhance bioavailability, serving as food or a food supplement. In the EU, this is allowed for several species such as *A.* protothecoides, *C. sorokiniana*, *C. vulgaris*, and *Parachlorella kessleri*, among others like *Scenedesmus vacuolatus*. Other species, such as *C. reinhardtii*, are in the pipeline of novel food regulation, while *Desmodesmus* species and other *Scenedesmus* species are currently not allowed as food.

Using random mutagenesis on permitted microalgal species with a history of safe consumption before 1997, such as *Chlorella*, poses no issues for regulatory approval. However, microalgae approved under the novel food regulation would require a new application under this regulation if random mutagenesis is used. Furthermore, the use of GMO could be controversial and face stricter regulations requiring careful evaluation and approval. Due to the relatively strict regulatory framework for GMOs in the EU, most commercial applications of gene editing technologies, including transgenic microalgae, have occurred outside the EU.

### 6.2. Comparing the Optimal Cultivation Strategies for Lutein Production

As seen from Table 3 and Table 4, employing fed-batch heterotrophic cultivation in a controlled fermenter results in the highest lutein productivity. Two-stage strategies, transitioning from heterotrophic to mixotrophic or photoautotrophic cultivation conditions, often result in slower lutein production and lower biomass density in the second stage. This contributes to lower final lutein concentrations in the PBR compared to those achieved with fed-batch heterotrophic conditions. Nevertheless, the lutein content is higher in these two-way strategies, which might be important for further downstream processing. Microalgal biomass with higher lutein content may facilitate greater extraction efficiencies during downstream processing, owing to a higher concentration gradient that enhances the diffusion of lutein from the cell interior to the extraction solvent. Consequently, achieving high lutein productivity from microalgae initially cultivated under heterotrophic conditions, which typically have lower lutein content, may require increased resources such as chemicals, energy, and water. However, this hypothesis warrants further investigation, as no comparative studies were found in the literature.

Energy consumption for artificial lighting under mixotrophic and phototrophic conditions can significantly increase the production cost of microalgae, which is not the case for heterotrophic cultivation. Perez-López et al. (2014) found that replacing artificial lighting with sunlight reduces both environmental impact and cost, though it can also lower biomass productivity [81]. Interestingly, Dineshkumar et al. (2016) tested various light strategies (constant light intensity, and linearly and exponentially increasing light intensity) for *Chlorella minutissima* [71]. They found that a linear light strategy not only increased lutein productivity and lutein content (5.35 mg/L/day; 8.24 mg/g DW) compared to constant illumination at 260 µmol/m^2^/s (4.32 mg/L/day; 6.37 mg/g DW), but also reduced light energy consumption by 32%. Apart from the energy costs associated with lighting, mixotrophic and phototrophic conditions can reduce the ecological footprint by lowering CO_2_ emissions, thanks to their carbon sequestration capabilities, which is not the case for heterotrophic cultivation.

Scalability of the used strategy is also important for commercialization and reducing production costs of microalgae. Additionally, a single-stage process might be easier to operate compared to a two-way strategy. Jeon et al. (2014) demonstrated that fed-batch heterotrophic cultivation for lutein production is scalable to a commercial level [72]. They observed that lutein concentration remained consistent for *C. vulgaris* when scaling up from a 5 L batch (252.75 mg/L) to fed-batches in a 25 m^3^ system (260.55 mg/L) and further to fed-batches in a 240 m^3^ system (263.13 mg/L). Scaling the two-way strategy to commercial levels has not been attempted. As seen in Table 3 and Table 4, most research on microalgal lutein production is predominantly confined to laboratory conditions with volumes often limited to 1 L bottles, PBR, or small-scale fermenters. Scaling the two-way strategy could present several challenges. For example, if the culture becomes dense under heterotrophic conditions in the first stage, it will need to be diluted for efficient light penetration when transferred to the second stage under mixotrophic or photoautotrophic conditions. Diluting the culture in closed systems would require larger PBRs, which would significantly increase the capital expenditure (CAPEX). Suparmaniam et al. (2024) hypothesize that two-way strategies can be upscaled by first cultivating the microalgae heterotrophically in fed-batch mode until the early exponential phase [23]. In the second stage, the microalgal cultures would be transferred to an open raceway pond (ORP) system operated under mixotrophic conditions. However, using ORPs under mixotrophic conditions might be challenging due to the increased risk of contamination because of the presence of a carbon source, resulting in culture crashes. Operating in photoautotrophic conditions for the second stage, ensuring the carbon source is fully utilized before changing conditions, might be a possible solution to minimize contamination risks in ORPs.

Under heterotrophic and mixotrophic conditions, both microalgae and bacteria also compete for organic carbon, with bacteria often outnumbering microalgae due to their shorter doubling times. As a result, keeping anexic conditions through the scaling process will be important to avoid culture crashes or contamination issues, which could raise safety concerns for human consumption. Selecting lutein-producing species that can resist extreme environments, such as low-pH and high-salt conditions, might offer a solution for contamination during cultivation. These resilient species could enhance the stability and efficiency of large-scale lutein production systems. For example, Bermejo et al. (2018) demonstrated that the acidophilic eukaryotic microalga *Coccomyxa onubensis*, which can endure moderate salt stress and low pH (pH 2.5), can accumulate lutein under phototrophic conditions (140 µmol/m^2^/s) with the addition of 100 mM NaCl, achieving up to 6.7 mg/g DW [80]. However, this species grows much slower compared to *Chlorella* species (Table 3), resulting in lower lutein productivity (1.63 mg/L/day).

### 6.3. Comparing Microalgae and Marigold for Lutein Production

#### 6.3.1. Advantages of Microalgal Production for Lutein Compared to Marigold

Natural lutein is currently produced commercially from marigold flowers, specifically *Tagetes patula* or *Tagetes erecta*. However, several advantages have led to growing interest in using microalgae for lutein production (Figure 2). The lutein content in the dry petal powders of *T. erecta* and *T. patula* ranges from 0.829 to 27.946 g/kg and 0.597 to 12.31 g/kg, respectively, depending on environmental conditions, growth stages, and genetic variation [22]. As a result, the lutein content in marigold petals can be higher than in dried microalgae (Table 3 and Table 4). However, microalgae have a growth rate that is 5–10 times faster than that of higher plants, which can significantly increase their lutein productivity [22]. Moreover, microalgae can be cultivated year-round, unlike seasonal marigold flowers. Li et al. (2015) compared the lutein yield, assuming an available lutein content of 20 g/kg in dry petal powder and 5 g/kg in dry microalgal powder [22]. Because of the higher growth rates and year-round production of microalgal biomass, the annual lutein production rate can reach 350–750 kg/hectare, whereas for marigolds, it is approximately 120 kg/hectare. The marigold production is also labor-intensive and predominantly located in economically upcoming countries such as China, India, and some African nations, which are locations that are prone to climate change (extreme temperature, drought, and heavy rainfall), which could affect the marigold production. Li et al. (2015) also stated that a lutein content of at least 10 g/kg (1% DW) in microalgae is deemed essential for commercial viability [22]. As discussed in previous sections, specific *Chlorella* strains can achieve lutein content ranging from 4 to 11 mg/g DW, depending on the cultivation conditions and strategies used, indicating that this criterion can be met with optimized cultivation methods. Furthermore, advancements in random mutagenesis and metabolic engineering offer potential for surpassing these values significantly, as shown for the *C. zofingiensis* mutant (13.81 mg/g DW) described by Huang et al. (2018) [33].

Additionally, microalgal cultivation requires no arable land and 2–10 times less water compared to marigold flowers. While microalgae require more nitrogen and phosphorus—1.5 and 2 times more, respectively—they need 3.5 times less potassium than marigold flowers [22]. Using food waste hydrolysate and other side streams can serve as sustainable and cost-effective nutrient sources, further enhancing the sustainability of microalgal lutein production. For instance, Wang et al. (2020) tested *Chlorella* sp. GY-H4 mixotrophic cultivation using food waste hydrolysate supplemented with 20 g/L glucose in semi-continuous mode, achieving a lutein productivity of 10.5 mg/L/day and a lutein content of 8.9 mg/g DW [82]. Additionally, ricotta cheese whey and cane molasses have shown potential as culture media for growing *Chlorella* species [83,84]. Tran et al. (2014) explored cost reduction by recycling the cultivation medium; however, this approach was shown to decrease both biomass and lutein content [85]. It is essential to note that legislation will require side streams used for microalgal lutein production to meet food-grade standards to ensure safety and compliance with health regulations.

#### 6.3.2. Disadvantages of Microalgal Production for Lutein Compared to Marigold

The primary drawback of lutein production from microalgae is its energy-intensive process, particularly in the downstream processing (harvesting, cell disruption, extraction, and purification) of the biomass [86]. To remove the larger volume of water in microalgal culture, extracting microalgal suspension requires more energy than for marigold flowers [22]. Conventional techniques like centrifugation consume significant energy, whereas sedimentation is time-consuming and carries a risk of lutein degradation. Drying requires similar amounts of energy for both marigold flowers and microalgae. Cell disruption is often necessary to increase carotenoid yields but is more energy-intensive than crushing/powdering the marigold flowers due to microalgae’s smaller cell size (3–10 μm), which reduces disruption efficiency [6]. Due to this, the energy needed for microalgal cell disruption ranges from 33 to 530 MJ/kg, about 1000 times higher than the energy required for crushing marigold flowers (800 kJ/kg) [22]. The composition, thickness, and size of microalgae cell walls dictate the energy demand for disruption. Lutein-rich *Chlorella* species, known for their strong cell walls, typically require cell disruption. In contrast, species such as *C. reinhardtii*, with more fragile cell walls, do not require cell disruption [87].

Despite microalgae requiring less water during cultivation compared to marigold flowers, the extraction phase for microalgae demands more solvents, water, and energy, attributed to stronger lutein bonds within the microalgal biomass and its small size [22]. Microwave-assisted extraction and supercritical fluid extraction are explored as potential alternatives for traditional solvent extraction methods due to their thermal stability and high efficiency, despite higher operational costs and energy demands [15]. Furthermore, an intense purification step is needed to eliminate water, chlorophyll, and other compounds bound to the free lutein in the microalgal cells. Advanced methods like chromatography offer high purification efficiency but are costly and challenging to scale up [15]. In contrast, marigold flower extracts contain primarily lutein and zeaxanthin esters, simplifying the extraction and purification processes. Typically, solvent extraction with n-hexane is employed to extract oleo-resin from milled dry flower petals, occasionally aided by KOH to release free lutein. For microalgae, solvent extraction, purification, and residual water removal for lutein purification are more laborious, with excess solvent requirements [24]. Balancing effectiveness with affordability remains a challenge in optimizing microalgal lutein extraction and purification processes.

Lutein extracted from marigolds can be marketed as an additive (E161b) in the European Union. Furthermore, it is granted GRAS status by the United States Food and Drug Administration for use as a health supplement promoting eye health [22]. In contrast, commercialization of purified lutein extracted from microalgal species would require regulatory approval. Alternatively, microalgae species rich in lutein can be used as a whole, bypassing the need for extraction and purification steps. However, consumer acceptance of this approach may be hindered by the distinct flavor and color characteristics of microalgae [88]. While marigolds are still the main source, research is ongoing to improve the efficiency and cost-effectiveness of microalgae for lutein production. In the future, we might see a shift towards microalgae as the preferred method.

## 7. Conclusions

The review highlights the potential and challenges of enhancing lutein production in microalgae through random mutagenesis and metabolic engineering. Both techniques have shown promise, with random mutagenesis achieving higher lutein content due to its application to naturally high-producing species, while metabolic engineering offers precision in modifying specific genes. The highest lutein content was observed in a *C. zofingiensis* mutant under stress conditions, yet scalability and productivity comparisons require further research. Regulatory considerations must be taken into account when using certain microalgae species, especially when employing genetic modification techniques.

Optimal cultivation strategies suggest fed-batch heterotrophic cultivation in controlled fermenters for the highest lutein productivity, though two-stage strategies, involving a transition from heterotrophic to mixotrophic or photoautotrophic conditions, show potential for higher lutein content, particularly in *Chlorella* species. The scalability of these cultivation strategies remains a challenge, with heterotrophic conditions being easier to scale than mixotrophic or photoautotrophic conditions. The review compares microalgal and marigold lutein production, highlighting microalgae’s faster growth rates and year-round cultivation advantages. Despite the current challenge of energy-intensive downstream processing in microalgae, advancements in extraction and purification technologies hold promise for overcoming these hurdles.

In conclusion, microalgae offer promising avenues for sustainable lutein production, particularly with advancements in genetic techniques for obtaining high-producing lutein species. Testing these species with the proposed cultivation strategies and scaling up the cultivation process are crucial for commercial viability. With ongoing research focused on optimizing cultivation and processing methods, microalgae have the potential to surpass marigolds as the preferred source of lutein in the future.

## Figures and Tables

**Figure 1 marinedrugs-22-00329-f001:**
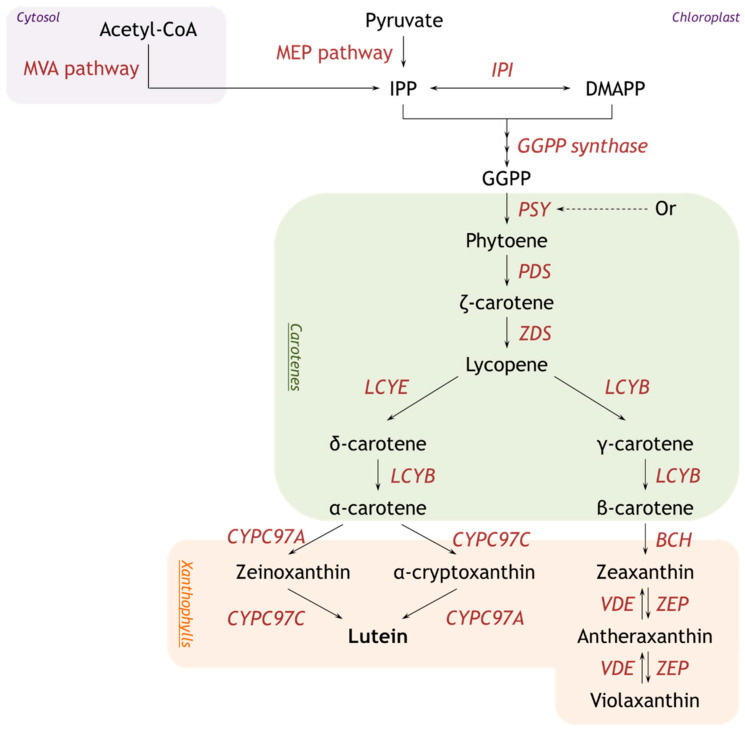
Schematic overview of the carotenoid biosynthesis pathway. The carotenoid biosynthesis starts with the synthesis of isopentyl pyrophosphate (IPP) through either the mevalonate (MVA) pathway or the methyl-D-erythritol phosphate (MEP) pathway located in the chloroplast [40]. IPP isomerizes to dimethylallyl diphosphate (DMAPP) and together they produce the immediate precursor of the carotenoid synthesis, geranylgeranyl pyrophosphate (GGPP). The condensation of two GGPP molecules leads to the formation of a colorless carotene, phytoene, by enzyme phytoene synthase (PSY). This PSY-catalyzed reaction is known to be the most important rate-limiting step in the carotenoid biosynthetic pathway [41]. Phytoene is converted to lycopene via a multi-step desaturation reaction catalyzed by phytoene desaturase (PDS) and ζ-carotene desaturase (ZDS) [42]. The cyclization of lycopene is the branching point of the carotenoid synthesis pathway into an α-branch and β-branch. In the α-branch, lycopene-ε-cyclase (LCYE) and lycopene β-cyclase (LCYB) catalyze lycopene to α-carotene, and then cytochrome P450 β-hydroxylase (CYPC97A) and cytochrome P450 ε-hydroxylase (CYPC97C) convert α-carotene to lutein [40]. In the β-branch, lycopene β-cyclase (LCYB) catalyzes lycopene to β-carotene, which is further hydroxylated by β-carotene hydroxylase (BCH) to form zeaxanthin [7]. Zeaxanthin epoxidase (ZEP) then converts zeaxanthin to violaxanthin. Violaxanthin can be converted to neoxanthin or enter the xanthophyll cycle, where it can be reversibly converted back to zeaxanthin via antheraxanthin to balance light harvesting and photoprotection.

**Figure 2 marinedrugs-22-00329-f002:**
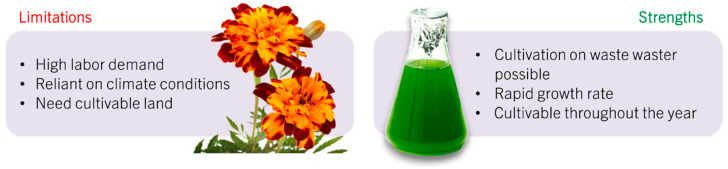
The advantages of using microalgae as a source for lutein production.

**Table 1 marinedrugs-22-00329-t001:** Health benefits of six carotenoids confirmed by human studies, their natural sources, and recommended dose (modified from Gong and Bassi, 2016 and Ren et al., 2021) [6,7].

Carotenoid	Health Benefits	Natural Sources	Recommended Dose
Astaxanthin	Strong anti-oxidant property	Shrimp; Salmon; Crabs; Microalgae (*Haematococcus* *pluvialis*) Phaffia rhodozyma	4–12 mg/day
	Anti-inflammatory effects		
	Anti-cancer		
	Cardiovascular health		
β-Carotene	Prevent night blindness	Pumpkin; Mango; Carrots; Microalgae (*Dunaliella salina*)	600 µg RE */day
	Anti-oxidant property		
	Prevents liver fibrosis		
Lutein	Prevents cataract and age-related	Marigold flower; Yolk; Broccoli; Microalgae; Orange-yellow fruits; Leafy green vegetables	6 mg/day
	macular degeneration		
	Anti-oxidant property		
	Anti-cancer		
	Prevents cardiovascular diseases		
Zeaxanthin	Anti-cancer	Marigold flower; Maize; Orange peppers; Microalgae; Scallions	2 mg/day
	Anti-inflammatory		
	Anti-allergy		
	Against UV, skin redness		
Fucoxanthin	Anti-obesity	Macroalgae; Microalgae	/
	Anti-oxidant property		

* RE, retinol equivalent.

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
