# Peer review of "Genetic Engineering and Innovative Cultivation Strategies for Enhancing the Lutein Production in Microalgae"

_marinedrugs, 2024, doi:10.3390/md22080329_

Round 1

Reviewer 1 Report

Comments and Suggestions for Authors

This manuscript describes genetic and metabolic engineering for production of lutein in microalgae. It is well organized and written; however, some points should be addressed for publication in Marine Drugs.

In addition, PubMed search of "lutein microalgae" lists 39 reviews, and a more general search of "carotenoids microalgae" lists 274, and 22 are published this year. Many of these are already in-depth reviews, demanding higher qualifications for publication.

Table 1 is the key summary table, and should thus include more details: main sources (both plants and microalgae, and animals if applicable), advantages (and disadvantages), and so on. And since authors described market values in the text, which can be included in the table. It is also suggested more details for lutein including cultivation conditions for specific sources for the best yield.

MVA pathways is not present in all microalgae, particularly the green algal lineages. Diatoms are exceptionally dual for MVA and MEP for carotenoid biosynthesis, which is missing in the manuscript. It is suggested that at least one section of text and some example entries in Table 2. Please discuss possible introduction of MVA pathway into green algae.

Additional points:

- Title number 4 is missing?

Author Response

Comment 1: This manuscript describes genetic and metabolic engineering for production of lutein in microalgae. It is well organized and written; however, some points should be addressed for publication in Marine Drugs.

Response 1: We would like to thank the reviewer for taking time to review our manuscript and suggest improvements. All the points are addressed below.

Comment 2: In addition, PubMed search of "lutein microalgae" lists 39 reviews, and a more general search of "carotenoids microalgae" lists 274, and 22 are published this year. Many of these are already in-depth reviews, demanding higher qualifications for publication.

Response 2: We acknowledge the extensive body of research on microalgal carotenoids. Before preparing this review, we conducted an exhaustive search of existing and qualitative reviews specifically addressing carotenoids and lutein. Our review distinguishes itself by exclusively focusing on strain improvement and cultivation strategies aimed at enhancing lutein content and lutein productivity. This targeted approach allows for a more detailed and precise analysis of lutein, avoiding broad generalizations common in reviews that cover a wide range of carotenoids. Unlike other reviews, we don’t delve into downstream processing or storage of carotenoids.

Furthermore, we include both random mutagenesis and metabolic engineering in our review. This is a rapidly evolving topic, and random mutagenesis is often excluded from previous reviews, making our inclusion of it particularly significant.

Unlike most reviews such as those by Fu et al. (2023) and Zheng et al. (2022), which focus on cultivation parameters, we focus on cultivation strategies, specifically fed-batch, batch, and semi-continuous methods. While most reviews focus on lutein content, our review also includes lutein productivity of the strategy, which is crucial for determining economic viability. We also provide a comparative analysis of lutein content and productivity across various microalgal species. Finally, we concentrate on studies conducted in photobioreactors (PBRs) larger than 1L, rather than small-scale flasks containing 100-150mL of culture.

Following text was likely added to the introduction to clearly define the scope and focus of the review. This addition aims to highlight the specific insights and depth of analysis that readers can expect from this review compared to others in the field.

“This review exclusively focuses on enhancing lutein content and productivity through strain improvement via random mutagenesis and metabolic engineering, alongside cultivation strategies such as fed-batch, batch, and semi-continuous methods. Unlike reviews that emphasize cultivation parameters for increasing carotenoid content broadly, our review specifically concentrates on lutein and offers a comparative analysis across various microalgal species.”

Comment 3: Table 1 is the key summary table, and should thus include more details: main sources (both plants and microalgae, and animals if applicable), advantages (and disadvantages), and so on. And since authors described market values in the text, which can be included in the table. It is also suggested more details for lutein including cultivation conditions for specific sources for the best yield.

Response 3: Table 1 in our review paper does not serve as the key summary table. Its primary purpose is to illustrate the range of carotenoids and their associated health benefits. We have included the main sources for each carotenoid, encompassing plants and microalgae, in Table 1. Additionally, we have incorporated the recommended doses for these carotenoids. While it would be valuable to provide a comprehensive list of advantages, disadvantages, and market values for each carotenoid and its sources, this would diverge from the specific focus of our study. As for the cultivation conditions for lutein, we have chosen not to include these in Table 1, as they are thoroughly discussed in detail later on in our review.

Comment 4: MVA pathways is not present in all microalgae, particularly the green algal lineages. Diatoms are exceptionally dual for MVA and MEP for carotenoid biosynthesis, which is missing in the manuscript. It is suggested that at least one section of text and some example entries in Table 2. Please discuss possible introduction of MVA pathway into green algae.

Response 4: Yes, indeed, the MVA pathway is not universally present in all microalgae; however, there are some microalgae (not discussed in the review) that contain the MVA pathway, such as Euglenophyta. Therefore, both the MVA and MEP pathways are depicted in the figure. A remark will be made in the review that the MEP pathway is more commonly found in green microalgae and is therefore more common for the synthesis of IPP. While there are diatoms, such as Phaeodactylum, that have an overall high carotenoid content, especially fucoxanthin, the lutein content is generally lower than that of green microalgae. As the review focuses on lutein production and there are not many publications in which random mutagenesis or metabolic engineering are used to increase lutein content in diatoms, they are not mentioned in the review/table.

Comment 5: Additional points: - Title number 4 is missing?

Response 5: Thank you for pointing out the issue with the numbering in the manuscript. We have realized that title number 3 was used twice by mistake. We corrected the numbering to ensure that all sections are properly labelled.

Reviewer 2 Report

Comments and Suggestions for Authors

The review higlights different cultivation strategies, just as genetic engineering techniques for incresing the production of lutein in microalgae. This is an outstanding work where the authors include many different data of the topic. However, I have some suggestions that could improve the value of the manuscript.

1. Line 71-72. The sentence is not clear. Please, rephrase it.

2. Line 114-115: Primary carotenoids mentioned in the paper are b-carotene, lutein and violaxanthin, which are typical carotenoids from green algae. Are there other primary carotenoids that are no typical for green microalgae, such as fucoxanthin or alloxanthin? If there is, please include it. 

3. Line 342-344: The authors said that cold temperatures slow down microalgal growth and lutein productivity. However, other studies using cold acclimated microalgae showed that this sentence is not completely true (https://doi.org/10.1007/s00449-023-02964-4; https://doi.org/10.1016/j.nbt.2022.12.001 ). It would be good if the authors discuss these exceptions.

Line 355: Change Table 3.

Line 585: The sentece looks weird. Please rephrase it.

Author Response

Comment 1: The review highlights different cultivation strategies, just as genetic engineering techniques for increasing the production of lutein in microalgae. This is an outstanding work where the authors include many different data of the topic. However, I have some suggestions that could improve the value of the manuscript.

Response 1: We would like to thank the reviewer for their valuable feedback and suggestions to improve our manuscript. All the reviewer's comments are addressed below. 

Comment 2: Line 71-72. The sentence is not clear. Please, rephrase it.

Response 2: We have rephrased the sentence. Before correction: As a result, the applications of synthetic carotenoids are mainly limited to animal feed and colorants. In contrast, natural carotenoids are generally considered safe and of higher customer preference for use in medicine or as supplements [10]. After correction: “Synthetic carotenoids are predominantly used in applications such as animal feed and as colorants, whereas natural carotenoids are preferred for use in medicine and food supplements [10]".

Comment 3: Line 114-115: Primary carotenoids mentioned in the paper are b-carotene, lutein and violaxanthin, which are typical carotenoids from green algae. Are there other primary carotenoids that are no typical for green microalgae, such as fucoxanthin or alloxanthin? If there is, please include it.

Response 3: We agree. There are other primary carotenoids that are not typical for green microalgae, such as fucoxanthin and alloxanthin. Fucoxanthin is a prominent carotenoid found in brown macroalgae (seaweeds) and certain diatoms (a type of microalgae). Alloxanthin, on the other hand, is commonly found in red algae (macroalgae) and some dinoflagellates (a type of microalgae). Lycopene is also a primary carotenoid found in multiple microalgal species. More primary carotenoids were included in the text.

Comment 4: Line 342-344: The authors said that cold temperatures slow down microalgal growth and lutein productivity. However, other studies using cold acclimated microalgae showed that this sentence is not completely true (https://doi.org/10.1007/s00449-023-02964-4; https://doi.org/10.1016/j.nbt.2022.12.001 ). It would be good if the authors discuss these exceptions.

Response 4: We appreciate your observation regarding the statement that cold temperatures slow down microalgal growth and lutein productivity. While our statement was based on existing literature concerning lutein and investigated microalgae (Chlorella sp., Chlamydomonas sp., Desmosdesmus sp. and Scenedesmus sp.) for lutein production only, it is clear that caution should be exercised in generalizing these findings for all microalgae. We will incorporate a discussion of these exceptions into our manuscript to provide a more comprehensive overview of the impact of cold temperatures on microalgal growth and lutein productivity: “However, Léon-Vaz et al. (2023) demonstrated that certain microalgal species produce higher lutein content at elevated light intensity (500 µmol/m²/s) and 10°C, compared to lower light intensity (100 µmol/m²/s) and 20°C. Consequently, the optimal temperature for lutein production varies by species and must be determined experimentally.”

Comment 5: Line 355: Change Table 3.

Response 5: Thanks you, this comment was addressed. 

Comment 6: Line 585: The sentence looks weird. Please rephrase it.

Response 6: We have rephrased the sentence. Before: “They found that constant 1 g/L NaAc every 12h feeding strategy yielded higher lutein productivity (8.04 mg/L/d) compared to the gradient strategy (6.11 mg/L/d).” After: “They found that feeding a constant NaAc concentration of 1 g/L yielded higher lutein productivity (8.04 mg/L/d) compared to the gradient feeding of NaAc (6.11 mg/L/d).”